# Estimating Adult Stature Using Metatarsal Length in the Korean Population: A Cadaveric Study

**DOI:** 10.3390/ijerph192215124

**Published:** 2022-11-16

**Authors:** Jeong-Hyun Park, Mijeong Lee, Digud Kim, Hyung-Wook Kwon, Yu-Jin Choi, Kwang-Rak Park, Suyeon Park, Sa-Beom Park, Jaeho Cho

**Affiliations:** 1Department of Anatomy & Cell Biology, School of Medicine, Kangwon National University, Chuncheon 24341, Republic of Korea; 2Department of Anatomy, School of Medicine, Keimyung University, Daegu 42601, Republic of Korea; 3Department of Biostatistics, Soonchunhyang University Seoul Hospital, Seoul 04401, Republic of Korea; 4Center of Biohealth Convergence and Open Sharing System, Hongik University, Seoul 04066, Republic of Korea; 5Department of Orthopaedic Surgery, Chuncheon Sacred Heart Hospital, Hallym University, Chuncheon 24253, Republic of Korea

**Keywords:** forensic science, forensic anthropology population data, personal identification, stature estimation, metatarsal bone, linear regression, cadaveric study

## Abstract

This study aims to propose a regression equation for estimating stature in the Korean population using metatarsal bones from cadavers and to validate the appropriateness of the Korean-specific equation by comparing it to equations from other populations. A total of 81 adult formalin-fixed cadavers (51 males and 30 females) were evaluated. The first and second metatarsal bones’ physiological and maximal lengths were measured, and the cadaveric stature of the subjects was determined as the distance from the vertex to the plantar face of the heel. In all measurements, the correlation coefficient between real stature and metatarsal length was statistically significant (*p* < 0.001). Additionally, both sexes showed a correlation between stature and metatarsal bone length. For unknown sex, M1 (first metatarsal maximal length) showed the strongest association between stature and metatarsal length. The following is the appropriate regression equation: 1172.4913 + 7.3275M1 (R = 0.703). The current equation demonstrated a statistically significant appropriateness for the Korean population when compared to equations for other populations (*p* < 0.001). In conclusion, we proposed a Korean-specific regression equation for estimating stature using metatarsal length, and this formula may be more appropriate and useful in forensic science for the Korean population.

## 1. Introduction

As the number of deaths and disappearances due to natural or massive disasters rises, personal identification has emerged as one of the most critical processes. Because forensic anthropological population data is required when a human body is severely burned or mutilated, the items available for identification are severely limited [1]. Sex, age, weight, and stature are all factors that influence biological profile. Among these, stature has significant forensic value because it provides forensic scientists with useful information as identifying the remains of the unidentified victim [2].

Because the anatomical method for estimating stature has the limitation that skeletal remains must be preserved, the mathematical method calculated from the association between various body segments and individual stature is being used to develop an accurate and reliable stature prediction model using anthropological population data [3,4,5]. Based on appropriate statistical analysis, these regression equations for estimating stature can be derived from various body parts, particularly bone. Long bones have typically been used in research that seek to estimate stature from bones using regression equations. The limitation has been proposed due to the fact that the long bones are frequently discovered scattered and shattered in the actual forensic field, making exact measuring impossible [6,7,8]. As a result, useful regression equations for estimating adult stature based on foot bones have been proposed [6,9,10,11].

In cadavers of Euro- and Afro-Americans, Byers et al. [12] were the first to show a correlation between metatarsal length and stature and to calculate stature from metatarsal bones using the regression equations. Additionally, that formula has been regularly employed in the field of forensics for more than 20 years. Regression models based on the corresponding population according to population affinity and region have, however, been proposed for more accurate stature estimation using metatarsals from cadaver, because forensic anthropological population data may demonstrate disparities by population affinity and region [13,14]. Although the authors have previously shown a favorable correlation between the length of the metatarsals as determined by radiography and stature in the Korean population [15], an additional study was required to develop a formula for determining stature by measuring the actual metatarsal length using a cadaver, however, due to the constraint that metatarsal length through indirect assessment using radiography can differ from the actual length.

This purpose of this study is to propose a regression equation for estimating stature in the Korean population using metatarsal bone from cadavers and to validate the appropriateness of the Korean-specific equation by comparing it to equations from other populations.

## 2. Materials and Methods

All cadavers used in this study were donated through the donation program with consent for medical school education and research. Furthermore, our institution of Institutional Ethics Committee this approved this study (Institutional Review Board number: CHUNCHEON NON2020-006).

When the sample size was analyzed based on the prior reports from other populations [13,14], at least 62 subjects (satisfied with R^2^ value of 0.67–0.72) and 49 subjects (satisfied with R^2^ value of 0.61–0.76), respectively, were requested. This was assuming a significance level of 0.01, a power of 99%, and a dropout rate of 30%.

A total of 84 formalin-fixed cadavers were dissected for this study. Among those 84 cadavers, 81 were included in this study, with 3 cadavers excluded due to abnormal signs of trauma or surgery, obvious deformities, pathologic lesions, and damage that make morphology investigation difficult. Of the 81 cadavers dissected from adult formalin-fixed cadavers, 51 (63%) were males and 30 (37%) were females. The donors’ average age at death was 70.36 (range, 20–97) years (Figure 1).

Cadavers fixed in formalin were positioned supine. The individuals’ cadaveric stature was measured as the distance from the vertex to the plantar face of the heel, and this value was measured with a sliding caliper. The lower limb was then secured by a pedestal to ensure foot stability. The length of the metatarsal bone was determined by measuring the first and second metatarsal bones of left side foot. To gain access to the bones, the skin and subcutaneous tissue, tendons, ligaments, and muscles were dissected, and the metatarsophalangeal joint and tarsometatarsal joint were disarticulated, and the first and second metatarsal bones were extracted from the cadaver’s foot. The following was the definition of metatarsal length measurement according to the previous study [14] (Figure 2).

F1 (physiological length of the first metatarsal): the distance between the most distal point of the head and the deepest point of the proximal articular surface in the first metatarsal bone.

M1 (maximum length of first metatarsal): the distance between the most distal point of the head and the tip of the tuberosity in the first metatarsal bone.

F2 (physiological length of the second metatarsal): the distance between the most distal point of the head and the deepest point of the proximal articular surface in the second metatarsal bone.

M2 (maximum length of second metatarsal): the distance between the most distal point of the head and the proximal tip in the second metatarsal bone.

All measurements were taken with an electronic digital caliper (Sincon Corporation, Seoul, Korea) with an accuracy of up to 0.1 mm. For all measurements, two researchers measured the length independently, and one researcher measured twice a week, with the average used as the measurement value.

The intraclass correlation coefficient was used to calculate inter- and intra-observer reliabilities for all measurements (ICC). According to the definition of Koo and Li [16], ICCs of greater than 0.90, between 0.75 and 0.90, between 0.50 and 0.75, and less than 0.50 were interpreted as excellent, good, moderate, and poor, respectively. The data analysis included means (m), standard deviations (SD), the correlation coefficient (R), standard error of estimate (SEE), adjusted determination coefficient (adj R2), root mean square error (RMSE) and Linear Regression Model (LRM). To estimate the relationship between the actual value and each predicted value, Spearman correlation analysis was used. To determine whether the difference between the actual and predicted values is zero, the Wilcoxon signed-rank test was used. Since our model results could be overestimated, we further evaluated the accuracy of our model using mean squared error (MSE) values and cross-validation methods. MSE was calculated in all formulas as a method of measuring the mean square difference between the estimated value and the actual value. That is, it was used to evaluate the accuracy in each formula. Additionally, to identify the interval validation in proposed formula, we performed the leave-one-out cross-validation (LOOCV) where the number of folds equals the number of instances in the data set. Thus, the machine learning algorithm is applied once for each instance, using all other instances as a training set and using the selected instance as a single-item test set. A *p* value < 0.05 was considered to indicate a significant difference. Data processing and statistical analyses were performed by R version 3.3.1 and Rex software (http://rexsoft.org/ accessed on 2 November 2022).

## 3. Results

The intraclass correlation coefficient for all measurements generated a result of 0.9 or higher. All measurements were interpreted as excellent and were employed in the study. As a result of the normality test for stature, the normality was satisfied as 0.855 in the female and 0.693 in the male, and 0.052 when both were considered.

Table 1 showed the descriptive analysis of male and female measurements in both groups. Males (*n* = 51) have an average age of 68.12 ± 15.33 years, which is less than females (*n* = 30) age, 74.17 ± 19.13 years. The stature differed significantly for corresponding male–female values (*p* < 0.001) by independent samples *t*-test. The average stature was found to be about 142.22 mm greater in males than females.

As summarized in Table 2, Table 3 and Table 4, the coefficient of correlation between the actual stature and metatarsal length was statistically significant in all measurements (*p* < 0.001). Additionally, the stature was correlated with the length of the metatarsal bone in both sexes. However, stronger correlations were detected between the stature and the metatarsal bone lengths in female. The highest correlation between the stature the metatarsal length was M1 (first metatarsal maximal length) for unknown sex. The corresponding regression equation is as follows: 1172.4913 + 7.3275M1 (R = 0.703)

The predicted stature value (PV) calculated by substituting the metatarsal length into the equations derived from this study and those derived from other population affinities was compared with the true stature value to validate the appropriateness of the Korean-specific equation (TV). All PV calculated using Korean-specific equations did not differ statistically from TV (*p* > 0.05) However, when the PV calculated by the equation derived from other races [13,14] was compared to the TV by corresponding sample, all showed statistically significant differences (*p* < 0.001) with the exception of the equation by Byers et al. [12] (Figure 3).

To validate the appropriateness of the Korean-specific equation by comparing it to equations from other populations, intra- and inter-groups validation analysis were performed. When estimating the actual stature of the Korean population using previously published formulas from other populations, the mean square error (MSE) of all measurements using our formula was significantly lower than that of other formulas, regardless of sex. Additionally, MSE was significantly lower than the cross validation error (CVE) values in all measurement, regardless of sex, as a result of leave-one-out cross-validation (LOOCV) (Table 5).

## 4. Discussion

Although a formula for estimating the stature by directly measuring the metatarsal length in cadavers of various population affinities has been proposed, the current study’s contribution is to provide forensic anthropological population data for the Korean population. In other words, the current study proposes a Korean-specific formula for estimating stature that can be applied appropriately to Korean populations by measuring the actual metatarsal length in a cadaver while taking into account morphological variability according to population affinity and region.

The anatomical technique can be used in forensic science to directly measure the stature of skeletal remains, but the skeletal remains must be preserved [3]. A precise equation created using a mathematical technique that closely correlates stature with a single bone has the advantage of being able to account for partial skeletal remnants. Additionally, small bones that are closely associated with stature are more valuable than long bones since they are less likely to be preserved [4,5]. Although European studies have been reported on estimating stature by measuring various measurements of the entire foot or footprint [17,18], the metatarsal bone is one of the best-preserved and complete bones in the forensic field, and it is unaffected by weight, which can significantly affect the estimation of stature among foot factors [19,20]. As a result, the current study is thought to be important in terms of proposing an equation for estimating stature using actual metatarsal length measurements obtained through dissection of larger samples.

In various population groups, equations for estimating stature using actual metatarsal length have been reported [12,13,14]. While Byers et al. [12] and Cordeiro et al. [14] found moderate relationships with correlation coefficients (respectively, maximum value 0.87 and 0.79), Bidmos et al. [13] and our study found slightly lower values (respectively, maximum value 0.73 and 0.70). The second metatarsal bone showed the strongest correlation between stature and metatarsal length in Western and African populations, but the first metatarsal bone did in this study on Korean populations. Skeletal development, including the proportions of the various bones, is influenced by a variety of factors, including population affinity and geographic location [21]. It also implies that factors such as population affinity and region have a direct impact on the regression equation for estimating stature. In current study, there was a significant difference between the estimated stature of the Korean population and the actual stature value when using the stature estimation formula derived from other races. As a result, in order to improve the accuracy of stature estimation using metatarsal length, it would be reasonable to develop an equation that is appropriate for each population.

A validation analysis was also performed to verify the Korean-specific formula derived from this study. The error (MSE) of our formula was significantly lower when compared to the estimation of actual stature of Koreans by substituting the stature estimation formula using the length of metatarsal bones extracted from cadavers for different population groups. Furthermore, even in the Korean population, the error value (CVE) of the corresponding formula in estimating actual stature was low. As a result, this study is significant in that it provides a Korean-specific regression equation for estimating the stature using metatarsal length that is applicable to Korean population.

Significant sex differences in stature and foot measurements were discovered [18]. This implies that the relationship between stature and foot measurements varies according to sex. Looking at the root mean square error (RSME) value of the regression formula in the results of this study, there is a difference between male and female, and it shows a lower value when sex is distinguished, indicating the need to calculate and represent the regression equation separately for each sex. A regression equation when sex is unknown is also presented, which can be used in forensic field that do not discriminate between sexes.

Although the authors demonstrated the correlation between the stature and metatarsal length in the Korean population using radiographically measuring the length of the metatarsal bone in living Koreans, the stature estimation formula using the radiographically measured metatarsal length had a limitation in its practical application because the maximum correlation coefficient (R) was slightly low at 0.48 [15]. The recent study suggested a novel equation for calculating stature that was derived by taking the real metatarsal bone from a cadaver and measuring it directly, because the metatarsal bone is unaffected by weight-bearing [19,20] The comparison between antemortem stature and the stature measured from cadaver showed no statistically significant difference [22], despite the fact that it may be debatable to assess the stature from a cadaver whose antemortem stature is unknown. As a result, it appears that the method of estimating Korean population stature by substituting metatarsal length using the new regression equation presented in this study is the most appropriate at the moment.

The biologic profile of a living person, such as age, sex and weight, can influence the estimation of stature. Although all of the subjects in this study were over the age of 20, the majority of the elderly adult cadavers were included due to the nature of the cadaver study. As a result, the possibility of a decrease in human stature due to aging and the postmortem change cannot be completely ruled out. Furthermore, because our sample contains a relatively small number of women, additional analysis by sex will be required after increasing the size of the female sample.

## 5. Conclusions

We proposed a Korean-specific regression equation for estimating stature using metatarsal length, which may be more appropriate and useful in forensic or legal practice for the Korean population. The maximum length of the first metatarsal bone (M1) has the strongest correlation with stature, with the following regression equation is as follows: 1172.4913 + 7.3275M1 (R = 0.703).

## Figures and Tables

**Figure 1 ijerph-19-15124-f001:**
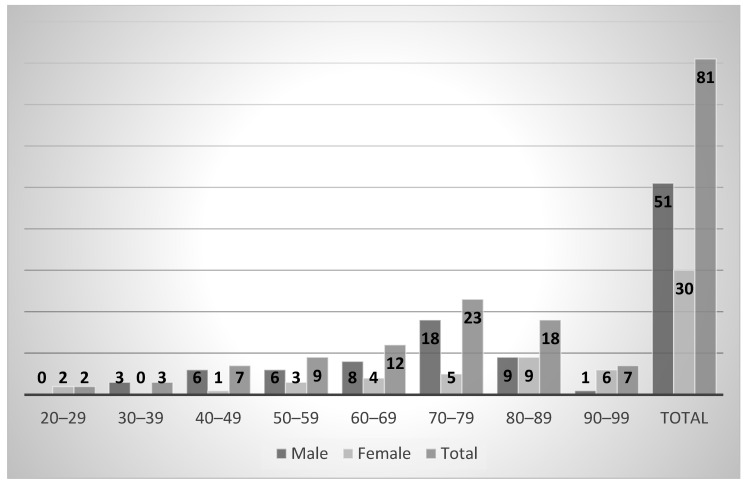
Sex and age distribution of Korean cadavers. (*n* = 81).

**Figure 2 ijerph-19-15124-f002:**
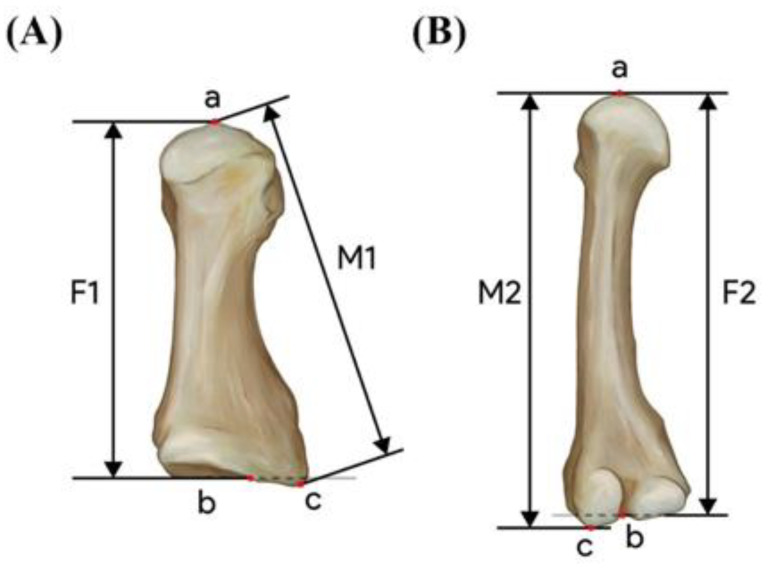
The measurements of metatarsal length. Metatarsal bones were observed superolaterally. (**A**) first metatarsal bone, (**B**) second metatarsal bone. F1, first metatarsal physiological length; F2, second metatarsal physiological length; M1, first metatarsal maximal length; M2, second metatarsal maximal length.

**Figure 3 ijerph-19-15124-f003:**
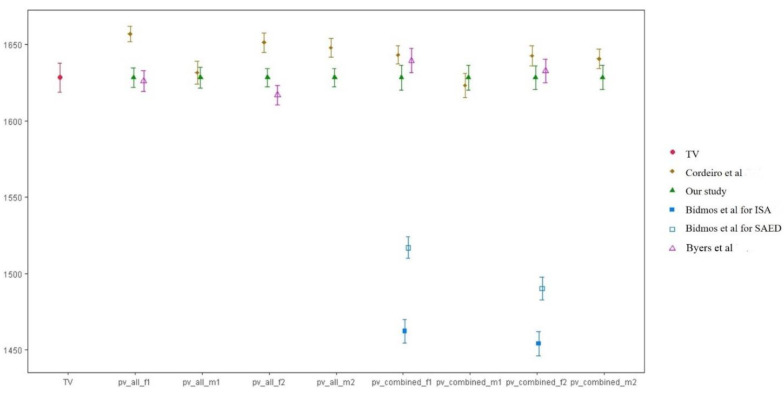
Comparison of true stature value (TV) with the predicted stature values (PV) by ours and formulas derived from other population affinities (Y-axis, stature (mm); X-axis, each stature value). TV, true stature value; PV, predicted stature value); F1, first metatarsal physiological length; F2, second metatarsal physiological length; M1, first metatarsal maximal length; M2, second metatarsal maximal length; all, unknown sex; combined, males and females combined; ISA, indigenous South African; SAED, South Africans of European descent. Cordeiro, et al. [14], Bidmos, et al. [13], Byers, et al. [12].

**Table 1 ijerph-19-15124-t001:** Summary statistics (Mean, standard deviation, median and range) of age, stature, F1 (first metatarsal physiological length, M1 (first metatarsal maximum length), F2 (second metatarsal physiological length) and M2 (second metatarsal maximum length) measurements (mm) of both sexes.

	Male (*n* = 51)			Female (*n* = 30)			Total (*n* = 81)		
	Mean ± SD	Median (IQR)	Range	Mean ± SD	Median (IQR)	Range	Mean ± SD	Median (IQR)	Range
Age	68.12 ± 15.33	60.45 (58.7, 62.85)	38 to 97	74.17 ± 19.13	80 (66.5, 86)	20 to 93	70.36 ± 16.97	75 (60, 82)	20 to 97
Stature	1681.05 ± 51.31	1687.5 (1646.25, 1715)	1530 to 1805	1538.83 ± 53.34	1533.75 (1511.25, 1578.75)	1400 to 1650	1628.38 ± 86.32	1640 (1562.5, 1692.5)	1400 to 1805
F1	60.98 ± 3.33	63.85 (62.38, 65.83)	55.45 to 72.25	56.57 ± 3.03	56.85 (54.33, 58.79)	50.3 to 61.95	59.35 ± 3.85	59.26 (57.05, 61.65)	50.3 to 72.25
M1	64.29 ± 3.92	72.85 (70.53, 75.4)	58.5 to 84.3	58.68 ± 6.15	59.77 (56.92, 61.29)	30.65 to 65	62.22 ± 5.54	62.9 (60, 64.8)	30.65 to 84.3
F2	72.91 ± 4.34	74.75 (72.95, 76.95)	65.4 to 93.6	66.77 ± 3.14	67.33 (64.68, 68.69)	59 to 71.8	70.63 ± 4.93	70.65 (67.65, 74)	59 to 93.6
M2	75.16 ± 4.46	72 (56, 79.5)	67.6 to 96.7	69.3 ± 2.98	69.1 (67.14, 70.95)	63.75 to 75.25	72.99 ± 4.87	73.3 (69.25, 76)	63.75 to 96.7

F1, first metatarsal physiological length; F2, second metatarsal physiological length; IQR, Interquartile range; M1, first metatarsal maximal length; M2, second metatarsal maximal length; SD, standard deviation.

**Table 2 ijerph-19-15124-t002:** Regression formulae for all (unknown sex, *n* = 81) with its correlation coefficient (R), adjusted determination coefficient (Adj R^2^), standard error of estimate (SEE) and root mean square error (RMSE).

Formula	R	Adj R^2^	SSE	RMSE
S = 695.3899 + 15.7963F1	0.678	0.453	63.82	73.62
S = 1172.4913 + 7.3275M1	0.703	0.487	61.81	78.59
S = 856.1170 + 10.9332F2	0.524	0.386	57.29	73.23
S = 832.5603 + 10.9028M2	0.616	0.371	57.21	74.47

F1, first metatarsal physiological length; F2, second metatarsal physiological length; M1, first metatarsal maximal length; M2, second metatarsal maximal length; S, stature; in mm.

**Table 3 ijerph-19-15124-t003:** Regression formulae for males (*n* = 51) with its correlation coefficient (R), adjusted determination coefficient (Adj R^2^), standard error of estimate (SEE) and root mean square error (RMSE).

Formula	R	Adj R^2^	SSE	RMSE
S = 1264.982 + 6.8584F1	0.398	0.141	45.62	50.94
S = 1141.9122 + 8.4526M1	0.460	0.196	45.69	50.56
S = 1455.3554 + 3.0956F2	0.262	0.050	45.60	52.59
S = 1464.3474 + 2.8831M2	0.251	0.044	42.27	52.17

F1, first metatarsal physiological length; F2, second metatarsal physiological length; M1, first metatarsal maximal length; M2, second metatarsal maximal length; S, stature; in mm.

**Table 4 ijerph-19-15124-t004:** Regression formulae for females (*n* = 30) with its correlation coefficient (R), adjusted determination coefficient (Adj R^2^), standard error of estimate (SEE) and root mean square error (RMSE).

Formula	R	Adj R^2^	SSE	RMSE
S = 1034.7979 + 8.9462F1	0.542	0.268	47.54	48.21
S = 1067.0078 + 7.9589M1	0.510	0.234	46.01	49.66
S = 1117.2403 + 6.3143F2	0.372	0.108	46.40	49.66
S = 969.6267 + 8.2134M2	0.460	0.183	46.94	51.83

F1, first metatarsal physiological length; F2, second metatarsal physiological length; M1, first metatarsal maximal length; M2, second metatarsal maximal length; S, stature; in mm.

**Table 5 ijerph-19-15124-t005:** Mean square error (MSE) for comparison between population and cross validation error (CVE) for interval validation of total sample in Korean population).

	Mean Square Error (MSE)	Cross Validation Error (CVE)
Our_all_F1	63.04	64.53
Cordeiro et al. _all_F1 [14]	72.54	
Our_all_M1	61.05	62.64
Cordeiro et al._all_M1 [14]	75.70	
Byers al_all_M1 [12]	78.40	
Our_all_F2	56.58	58.04
Cordeiro et al._all_F2 [14]	67.05	
Our_all_M2	56.51	57.82
Cordeiro et al._all_M2 [14]	67.60	
Byers al for_all_M2 [12]	73.89	
Our_male_F1	46.60	48.68
Cordeiro et al._male_F1 [14]	50.80	
Our_male_M1	45.10	47.34
Cordeiro et al._male_M1 [14]	50.52	
Byers al for_male_M1 [12]	50.52	
Bidmos et al. for ISA_male_M1 [13]	50.52	
Bidmos et al. for SAED_male_M1 [13]	50.52	
Our_male_F2	45.48	54.01
Cordeiro et al._male_F2 [14]	50.80	
Our_male_M2	47.39	54.38
Cordeiro et al._male_M2 [14]	50.79	
Byers al for_male_M2 [12]	50.79	
Bidmos et al. for ISA_male_M2 [13]	50.79	
Bidmos et al. for SAED_male_M2 [13]	50.79	
Our_female_F1	44.07	48.65
Cordeiro et al._female_F1 [14]	48.20	
Our_female_M1	45.11	50.00
Cordeiro et al._female_M1 [14]	51.76	
Byers al for_female_M1 [12]	51.76	
Bidmos et al. for ISA_female_M1 [13]	51.76	
Bidmos et al. for SAED_female_M1 [13]	51.76	
Our_female_F2	44.06	48.52
Cordeiro et al._female_F2 [14]	49.60	
Our_female_M2	40.84	45.26
Cordeiro et al._female_M2 [14]	50.71	
Byers al for_female_M2 [12]	50.71	
Bidmos et al. for ISA_female_M2 [13]	50.71	
Bidmos et al. for SAED_female_M2 [13]	50.71	

F1, first metatarsal physiological length; F2, second metatarsal physiological length; M1, first metatarsal maximal length; M2, second metatarsal maximal length; all; males and females com-bined; ISA, indigenous South African; SAED, South Africans of European descent.

## Data Availability

The data used to support the findings of this study are available from the corresponding author upon request.

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
