# Peer review of "Estimating Adult Stature Using Metatarsal Length in the Korean Population: A Cadaveric Study"

_ijerph, 2022, doi:10.3390/ijerph192215124_

Round 1
Reviewer 1 Report (Previous Reviewer 1)
The authors have adequately incorporated all the suggestions, so I suggest that the article be published in Int. J. Environ. Res. Public Health.
Author Response
Thank you for comment about publication.
Reviewer 2 Report (Previous Reviewer 2)
Overall, the authors adequately addressed the peer review comments. I have just a few minor suggestions:
p. 2/10 ln. 56: "usefulness" should be "useful"
p. 2/10 ln. 63: "metatarsal bones" should be "metatarsals"
p. 8/10 lns. 251, 253, 255, and 257: "gender" and "genders" should be changed to "sex" and sexes" (make sure all "gender" is removed from the manuscript).
Author Response
Overall, the authors adequately addressed the peer review comments. I have just a few minor suggestions:
p. 2/10 ln. 56: "usefulness" should be "useful"
Thank you for comment. We revised as your comment.
p. 2/10 ln. 63: "metatarsal bones" should be "metatarsals"
Thank you for comment. We revised as your comment.
p. 8/10 lns. 251, 253, 255, and 257: "gender" and "genders" should be changed to "sex" and sexes" (make sure all "gender" is removed from the manuscript).
Thank you for comment. We revised as your comment.
Reviewer 3 Report (Previous Reviewer 3)
Dear Editor,
Thank you very much for having trusted me to review this manuscript. The revised version of manuscript is improved, the Authors incorporated the suggestions provided by the reviewers. I thank the Authors for the great effort they have made in responding to the comments of the reviewers. The manuscript is very interesting and can be accepted for publication. However, there are two errors that need attention:
*As I suggested in the first review, the terms “sex” and “gender” are not synonyms. Please, change the term "gender" to "sex" in lines 200, 201, 251, 253, 255, 257 and 258; they are not interchangeable.
*Lines 256-257. Change the sentence “A regression equation for all gender is also presented […]” to “A regression equation when sex is unknown is also presented”.
Author Response
Dear Editor,
Thank you very much for having trusted me to review this manuscript. The revised version of manuscript is improved, the Authors incorporated the suggestions provided by the reviewers. I thank the Authors for the great effort they have made in responding to the comments of the reviewers. The manuscript is very interesting and can be accepted for publication. However, there are two errors that need attention:
*As I suggested in the first review, the terms “sex” and “gender” are not synonyms. Please, change the term "gender" to "sex" in lines 200, 201, 251, 253, 255, 257 and 258; they are not interchangeable.
Thank you for comment. We revised as your comment.
*Lines 256-257. Change the sentence “A regression equation for all gender is also presented […]” to “A regression equation when sex is unknown is also presented”.
Thank you for comment. We revised as your comment.
This manuscript is a resubmission of an earlier submission. The following is a list of the peer review reports and author responses from that submission.
Round 1
Reviewer 1 Report
This article offers a contribution to the topic of the biological profile of individuals, in this case focusing on cadaveric stature. The idea of the manuscript is interesting and may be helpful to the forensic society. The manuscript is well organised, but it is missing some important information that I would like to have answered, as well as some important changes that need to be made before publication.
* The authors should distinguish between forensic stature, living stature, and cadaveric stature, and all of these terms should be discussed appropriately in both the introduction and the discussion.
* Most cadavers were 70-79 years old (Fig. 1). There are definitely significant changes in foot structure and stature. The age range of the sample presented in the article is too heterogeneous. The authors should discuss the age factor and associated changes, e.g., changes in foot structure and stature after the age of 30 years. This should also definitely be included as a limitation of the study, as it could skew the results in this particular study. It would be optimal to split the file in terms of age, but unfortunately the sample size is not large enough for this. At least if it is in this form with the current age range, it should be given more attention and discussed in more detail.
* In the Material and Method section, you mention that two researchers were involved in the measurement - so did you calculate both intra- and inter-observer error?
* Did you only measure on one foot? - Can you explain this in more detail? What about possible bilateral differences?
* Authors should comment on the normality distribution of the data when doing a regression analysis.
* I see that the independent t-test for sexual dimorphism was performed with significant results. However, I suggest testing whether sex has an effect on the relationship between stature and foot measurements (GLM).
* Please avoid the term "race" and use "ethnic variety" instead.
* The accuracy of the proposed formulas (calculating MD or RMSE for your sample size) needs to be discussed.
* Discussion of weight-bearing and non-weight-bearing measurements and the differences should be mentioned in both the introduction and the discussion section.
* European studies on foot, hand, and footprint measurements should also be mentioned in the introduction or discussion, e.g.:
- P. Uhrová, R. Beňuš, S. Masnicová, Stature estimation from various foot dimensions among Slovak population, J. Forensic Sci. 58 (2013) 448-451;
- Švábová et al. Estimation of stature and body weight from static and dynamic footprints - Forensic implications and validity of non-colouring cream method. Forensic Science International 330 (2022), 111105.
Reviewer 2 Report
General comments:
Overall, this article presents an equation to estimate stature based on the first and second metatarsals for a Korean cadaveric sample. While stature methods developed on multiple elements are necessary, I am unsure how common or beneficial stature estimation from the metatarsals would be compared to the long bones of the lower limb. Is there literature that demonstrates that foot elements are more frequently represented compared to larger long bones? Additionally, the authors use “gender,” “men,” and “women” when the terms should be “sexes,” “male,” and “female,” respectively, since at present forensic anthropologists cannot predict gender (psychological, cultural, self-identity) from the skeleton. Similarly, the authors use “race and ethnicity” as an explanation for population differences in stature; however, race is not biological and race does not sufficiently explain human skeletal variation. The biological consequences of racism could have effects on stature, but this article is not addressing the effects of racism. Therefore, I suggest using “population,” “population affinity,” or “ancestry” instead of “race and ethnicity.”
While Figure 3 presents the true values and estimated values in a visual way, I think a table that shows how well the metatarsal regression formulae performed on this sample would be useful. Did the regressions over- or underestimate stature? How often was the regression incorrect? Overall, what is the accuracy of this method. In other words, what percentage of cadavers were correctly assigned a stature based on the regression equation? Do you have Korean skeletons in which to validate this method? That is, skeletons that were not included in developing the method?
The citations are a little scant (n=20)—I think there has been other work in Asia to create population-specific stature methods, and other studies that have used metatarsals to predict stature (in other non-Asian countries). Relatedly, I think that the studies featured in Figure 3 (e.g., Bidmos et al., Cordeiro et al. etc.) should be explicitly discussed in the Introduction so that the reader understands what populations and methods were used in those studies. At present, we don’t know if they are comparable to the present study.
Moreover, I am curious what the authors think is driving the population specificity of this Korean sample? The authors offer some generalities with "race" and weight, but what specific to Korea would render the other stature regression equations useless in Korean contexts? I think this would be beneficial to address in the Discussion.
Below are more specific comments:
p. 1, ln. 28: “genders” should be “sexes” since the article is discussing females and males and anthropologists at present cannot estimate gender.
p. 1, ln. 40: Should be: “increasing natural and mass disasters.”
p. 1, ln. 43: “gender” should be “sex”
p. 1, ln. 45: Suggest removing “as evidence at the crime scene”—stature is not helpful in processing a crime scene.
p. 2, ln. 47: The anatomical method is often referred to as the “Fully” method that uses all bones that contribute directly to stature. Other statures are regressions—wherein they are estimated by measuring a single or couple bones. Suggest defining these here to clarify what you mean about the “anatomical method.”
p. 2, ln. 54-58: In my experience, foot and hand elements are often scattered or not recovered compared to the more robust and larger long bones, particularly of the lower limb. I don’t necessarily think that shoes will regularly protect the feet particularly in prolonged postmortem intervals.
p. 2, ln. 63: Suggest changing “race and ethnicity” to “population affinity” since race and ethnicity are not biologically based.
Figure 1: Suggest using “20-29” instead of “20~29”. Also, suggest putting the totaled numbers in white above each of the bars. Caption should read “Sex and age distribution of the Korean cadavers (n=81).”
p. 3, ln. 100: sentence should end with a : “…was as follows (Figure 2):”
p. 4, ln. 136: “Men” should be “Males”; “women” should be “females”
Table 1: What is “IQR”? This should be defined in the narrative.
p. 5, ln. 50: “genders” should be “sexes”
p. 6, ln. 170: “races” should be “population affinities” or “ancestries”
p. 6, ln. 174: “Byres” should be “Byers” I think
Figure 3: This figure lists several stature studies of various population affinities; however, these are not directly discussed in the Introduction. For better context, I suggest discussing what each of these studies is about (samples, methods, etc.). Are they all based on metatarsal length? Also, should read “South Africans of European Descent (SAED).”
p. 6, ln. 184: “races” should be “populations” “population affinities” or “ancestries.”
p. 6, ln. 183-189: While there are certainly population differences, why do Koreans differ from other populations in terms of stature? What are the driving forces that render the non-population-specific equations from working well on the Korean sample? I think this would be very helpful to address in the Discussion.
p. 7, ln. 194-196: Again, I think that smaller bones are less likely to be preserved in forensic contexts (e.g., femora, tibiae) compared to hand and foot elements.
p. 7, ln. 205: Bidmos et al. needs a citation number here.
p. 7, ln. 210: Race and ethnicity in and of themselves does not affect stature as they are not biologically based, but rather sociocultural constructions. There are population differences in stature, but it is more nuanced that race. For example, Bergman’s and Allen’s rules help to explain why some populations are taller or shorter than others—this is not race, but adaptations to differing environments and climates.
p. 7, ln. 230-238: How does weight impact stature? What is the relationship? Suggest getting into the literature to explain this.
Reviewer 3 Report
I reviewed the article entitled: “Adult stature estimation from metatarsal length in a Korean population: A cadaveric study”. The study analyzes the physiological and maximal lengths of first and second metatarsal bone from cadavers to generate specific equations to estimate stature in the Korean population.
The manuscript presents an interesting method of stature prediction, however contains some important errors that, due to this, make me not recommend the manuscript for publication in the International Journal of Environmental Research and Public Health.
Below are some appointments of the article:
*I advise the authors to find a native English speaker to proofread the manuscript.
*Line 42. Change “separated” to “mutilated” or “dismembered”.
*Line 43. In forensic research is not correct the use of the term “determination” of biological profile. Forensic anthropologists do not determine the biological profile (with an accuracy of 100%); they attempt to estimate or diagnose it (assumes an error range). This is an important distinction. I suggest replacing it using more adequate terms as “estimation” or “diagnosis”.
*Line 43. The variable “gender” is used as synonym of “sex”. Sex and gender are not synonyms. “Sex” is a biological term. It refers to the biological differences between males and females, such as the genitalia and genetic differences. “Gender” is more difficult to define, but it can refer to the role of a male or female in society (gender role) or an individual’s concept of themselves (gender identity). Please, change these terms as they are not interchangeable. Check all the manuscript.
*Line 63. *Whereas “ancestry” is a measurable biological parameter, “race” is the product of historical, social, and political processes and not a “natural” or biological division of human variation. Please, remove the term “race” using more adequate terms as “ancestry” or “human population”. Please, check all the manuscript.
*Figure 1. The numbers in Figure 1 are hard to see. Please use the color black on the numbers to increase the contrast with the background.
*Lines 120-123. For intra- and inter-observer error analysis the Intraclass Correlation Coefficient (ICC) was calculated. However, to determine the strength of agreement, the ICC calculated was compared to the criteria proposed by Landis and Koch (1977). These criteria were proposed for categorial data (qualitative data; using Kappa statistics), and they should not be used for continuous variables (quantitative data; using ICC). Please, use more adequate criteria to determine the strength of agreement using ICC. For example: Fleiss (1986) (J.L. Fleiss, Design and analysis of clinical experiments, John Wiley & Sons, Nueva York, 1986) or Koo and Li (2016) (https://doi.org/10.1016/j.jcm.2016.02.012).
*Material and methods. Which side of the foot was measured? Left or right metatarsals? More details must be given.
*Table 2. Table 2 shows the regression equations for pooled sexes (or combined sexes), and can be used to identify of and individual of unknown sex. When the sex of the skeletal individual is known, regression equations of Table 3 (for male sex) and Table 4 (for female sex) can be used. Please change Table 2 caption and text of lines 152-154.
*Why do the equations with the sexes combined (Table 2) have a better fit than the equations divided by sex (Tables 3 and 4)? These results mean the opposite of what one would expect. This situation should be discussed in the manuscript.
*This study aims to propose regression equations for estimating stature from Korean population and to verify the appropriateness of the Korean-specific equations by comparing with equations from other populations. However, there is a big problem with this approach. Authors use metatarsal bones from cadavers (fresh bones) and other authors, such as Byers et al. [12] and Bidmos [13], use metatarsal bones from osteological collections (dry bones). The length measurement of the bones varies according to the methodology used. Dry bones are shorter relative to fresh bones and require the implementation of corrective measure into regression formulas. Thus, the different measurements taken by the Authors cannot be used in the Byers and Bidmos equations.